# OpenReview forum: "Jigsaw Puzzles: Splitting Harmful Questions to Jailbreak Large Language Models"
_ICLR.cc/2025/Conference — ICLR 2025 Conference Withdrawn Submission_

### Official Review · Reviewer_YW3D · 2024-10-29

**Soundness:** 2
**Presentation:** 4
**Contribution:** 3
**Rating:** 5
**Confidence:** 5

**Summary:**

**Advantages**

The paper introduces a novel multi-turn jailbreak strategy termed Jigsaw Puzzles (JSP), designed to test and expose the vulnerabilities of Large Language Models (LLMs) when confronted with fractionated queries. JSP bypasses the inner guardrails of LLMs by segmenting harmful questions into harmless fragments, which are then concatenated together to induce the models to generate harmful content.

JSP demonstrates a high success rate of 93.76% across 189 harmful queries on five advanced LLMs, including Gemini-1.5-Pro, Llama-3.1-70B, GPT-4, GPT-4o, and GPT4o-mini, highlighting the safety risks inherent in multi-turn interactions with LLMs.

****
**Disadvantages**

The method is simple yet effective and is quite interpretable. However, In the experimental design, there are several shortcomings:

* I'm confused regarding the implementation of the "Standard Prompting" baseline in Table 1. It is ambiguous whether the baseline merely employs stage 1 for query rewriting or directly inputs the queries to the model without rewriting.

* The authors have only compared JSP with some single-turn attack methods (PAP and PAIR). Since JSP is inherently a multi-turn attack method, and multi-turn attacks are generally more effective than single-turn attacks, a fair comparison should be made with multi-turn attack baselines, such as those referenced in [1,2], to demonstrate the effectiveness of JSP. The authors are encouraged to include these comparisons to provide a more comprehensive assessment of JSP's capabilities.

* The robustness of JSP against existing jailbreak defense methods has not been adequately tested. Although the authors have tested JSP against two system-prompt-based defense approaches, these tests do not constitute universally recognized and effective defense baselines. Based on my understanding of JSP, I suggest the authors test JSP against the following three defenses:

  * Gradient Cuff [3]: The results show that ASR(a) is significantly lower than ASR(q), indicating that the model's responses to JSP queries still have high uncertainty (meaning sometimes it refuses and sometimes it does not). Thus, the authors need to compare the effectiveness of JSP in breaking through Gradient Cuff defenses (which defend against jailbreak attacks based on response uncertainty).

  * Self-Reminder [4]: While JSP can evade the built-in safety checks of LLMs, the Self-Reminder method focuses on restricting LLM outputs no matter the harmfulness of user inputs. Therefore, the authors need to test the effectiveness of JSP against defenses employing the Self-Reminder system prompt.

  *  Perplexity (PPL [5] ): JSP splits sentences and words, making user input less coherent. It would be interesting to see whether JSP can bypass defenses based on text perplexity detection.

  * Safe-Decoding [6]: Safe decoding utilizes a safety-aware expert LLM to assist in decoding; therefore, the authors should explore whether JSP methods can still induce LLMs to output harmful content when an external expert LLM controls the decoding process.

[3,5,6] required the LLMs to be open-source, so the authors only need to conduct tests on Llama-3.1-70B and report the results. [4] are expected to test on all LLMs.

****
**References**

[1] Multi-Turn Context Jailbreak Attack on Large Language Models From First Principles. Xiongtao Sun, Deyue Zhang, Dongdong Yang, Quanchen Zou, Hui Li.

[2] Many-shot Jailbreaking. Anthropic.

[3] Gradient Cuff: Detecting Jailbreak Attacks on Large Language Models by Exploring Refusal Loss Landscapes. Xiaomeng Hu, Pin-Yu Chen, Tsung-Yi Ho

[4] Defending ChatGPT against jailbreak attack via self-reminders. Yueqi Xie, Jingwei Yi, Jiawei Shao, Justin Curl, Lingjuan Lyu, Qifeng Chen, Xing Xie Fangzhao Wu.

[5] Baseline Defenses for Adversarial Attacks Against Aligned Language Models. Neel Jain, Avi Schwarzschild, Yuxin Wen, Gowthami Somepalli, John Kirchenbauer, Ping-yeh Chiang, Micah Goldblum, Aniruddha Saha, Jonas Geiping, Tom Goldstein.

[6] SafeDecoding: Defending against Jailbreak Attacks via Safety-Aware Decoding. Zhangchen Xu, Fengqing Jiang, Luyao Niu, Jinyuan Jia, Bill Yuchen Lin, Radha Poovendran.

**Strengths:**

**Strengths 1** The paper introduces a novel multi-turn jailbreak strategy termed Jigsaw Puzzles (JSP), designed to test and expose the vulnerabilities of Large Language Models (LLMs) when confronted with fractionated queries. JSP bypasses the inner guardrails of LLMs by segmenting harmful questions into harmless fragments, which are then concatenated together to induce the models to generate harmful content.

**Strengths 2** JSP demonstrates a high success rate of 93.76% across 189 harmful queries on five advanced LLMs, including Gemini-1.5-Pro, Llama-3.1-70B, GPT-4, GPT-4o, and GPT4o-mini, highlighting the safety risks inherent in multi-turn interactions with LLMs.

**Strengths 3**  The method is simple yet effective and is quite interpretable.

**Weaknesses:**

**Weaknesses 1** The authors have only compared JSP with some single-turn attack methods (PAP and PAIR). Since JSP is inherently a multi-turn attack method, and multi-turn attacks are generally more effective than single-turn attacks, a fair comparison should be made with multi-turn attack baselines.

**Weaknesses 2** The robustness of JSP against existing jailbreak defense methods has not been adequately tested. Although the authors have tested JSP against two system-prompt-based defense approaches, these tests do not constitute universally recognized and effective defense baselines.

**Questions:**

**Question 1**: I'm confused regarding the implementation of the "Standard Prompting" baseline in Table 1. It is ambiguous whether the baseline merely employs stage 1 for query rewriting or directly inputs the queries to the model without rewriting.

**Question 2**: Could the author compare JSP with some multi-turn jailbreak baselines? Such as [1] and [2].

**Question 3**: Is JSP robust against universally recognized and effective defense baselines? The author should evaluate JSP's effectiveness against the [3,4,5,6].

**Details Of Ethics Concerns:**

no concerns.

---

### Official Review · Reviewer_Qkeg · 2024-11-02

**Soundness:** 3
**Presentation:** 3
**Contribution:** 2
**Rating:** 3
**Confidence:** 4

**Summary:**

The paper introduces a multi-turn jailbreak method, Jigsaw Puzzles (JSP), which bypasses safeguards in LLMs by splitting harmful questions into harmless parts for reassembly, achieving a 93.76% success rate across five models and revealing a significant vulnerability.

**Strengths:**

1. The JSP strategy achieved an impressive average attack success rate of 93.76%, indicating its effectiveness in bypassing LLM safeguards.
2. Even under defence measures, JSP maintains a high success rate, showing its robustness against current defence strategies.
3. The paper provides a comprehensive analysis of the JSP strategy under various settings, including prompt design, splitting strategy, and turn settings.

**Weaknesses:**

1. The dataset evaluated in this study is somewhat limited, consisting of only 189 harmful questions. This sample size is relatively small, so evaluating on additional datasets would enhance the robustness of the findings.
2. Including other multi-turn jailbreak methods [1,2] for comparison would help demonstrate this method’s effectiveness more comprehensively.
3. The performance of this method in single-attempt shown in Figure 7 seems suboptimal, particularly in Gemini and GPT-4o. Further discussion and analysis on these limitations would be beneficial.
4. Although not necessary, it would be beneficial to include results for longer questions and languages beyond English. This would provide a more thorough assessment of the method’s applicability across varied linguistic contexts.

[1] Great, Now Write an Article About That: The Crescendo Multi-Turn LLM Jailbreak Attack https://arxiv.org/pdf/2404.01833

[2] Chain of Attack: a Semantic-Driven Contextual Multi-Turn attacker for LLM https://arxiv.org/pdf/2405.05610

**Questions:**

N.A.

---

### Official Review · Reviewer_dQJi · 2024-11-02

**Soundness:** 1
**Presentation:** 2
**Contribution:** 1
**Rating:** 3
**Confidence:** 5

**Summary:**

In this paper, the authors proposed JSP (JigSaw Puzzle) as a multi-turn jailbreaking method that features rewriting and splitting of a malicious request into multiple chunks that fills up a multi-turn dialog with the victim LLM to sidestep its built-in safety guardrails and defensive system prompts. The authors conducted experiments with different splitting choices, real and pseudo multi-turn setup, and two system defensive system prompts on GPT 4/4o Gemini 1.5 Pro and Llama 3.1 70b models using a subset of SafeBench to demonstrate JSP's effectiveness, merits of attacking in multiple turns, and resistance to defenses with no to limited amount of comparison with non-trivial single-/multi- turn baselines either performance-wise or efficiency-wise.

**Strengths:**

The authors introduced pseudo multi-round setup as a middle ground of single and (JSP-fashioned) multi-round jailbreaking methods, which is an interesting way to investigate the impact of conversational "special tokens" on safety, although the experiment is not solid or comprehensive enough to derive a convincing conclusion about it.

**Weaknesses:**

1. Lack of Novelty: The proposed method of attacking LLMs through multi-round conversations has been previously explored in works such as [1] and [5]. Similarly, the technique of splitting malicious inputs into chunks is not novel either.
2. Inadequate and Potentially Inappropriate Dataset Selection: The authors utilize SafeBench, originally designed by FigStep for jailbreaking vision-language models (VLMs), rather than LLMs. SafeBench does not demonstrably outperform or offer more comprehensive coverage compared to datasets like HarmBench, AdvBench, or Malicious-Instruct, which are commonly used for evaluating LLM jailbreaks. Choosing it provides no clear benefit and only makes it difficult to compare JSP with existing methods. Not to mention the authors have discarded 30% of the malicious topics from SafeBench out of no explicit reasons.
3. Limited and Potentially Unrepresentative Model Choices: Claude 3 models are popular alternatives to GPTs, and they are known to be safer than many LLMs including GPT 4  [4,5] due to its use of a model dedicated to notifying the underlying LLMs of potential jailbreaking attempts. It is interesting to see how JSP deals with them. Additionally, higher general capabilities of LLMs don't guarantee stronger resistance to attacks. Llama 2, for instance, is knowingly hard to jailbreak and the 7b variant is even harder than the 13/70b variants to many attacks [2,3]. Evaluating JSP on a more diverse set of models would provide a clearer picture of its effectiveness.
4. Insufficient Baseline Comparisons: The main experiments do not compare JSP with established attack methods, single- or multi-turn. Single-turn attacks put on stricter constraints than multi-turn attacks. Therefore, if JSP cannot significantly outperform single-turn attacks, it cannot justify its higher cost, longer latency and other drawbacks due to the multi-turn nature. The authors do have a small comparison with PAIR and PAP in section 5 using an AdvBench subset, but PAIR and PAP are not up-to-date or among the most powerful attacks. [3,4], for instance, have shown clear advantage over them. Multi-turn attacks [1,5] are not discussed at all in the paper. In fact, these work also investigated the difference between single- and multi-turn attacks, only more in-depth than this paper.
6. Superficial Defense Strategies: The handwritten system prompts can hardly represent the range of existing defenses. As the authors are well aware in Section 2, there are LLMs tuned specifically for safety such as [6] and exterior defensive mechanisms like input [7] and output filtering [8], yet they are not studied at all.
6. Confusing and Incomplete Evaluation Metrics: The paper introduces ASR-a alongside the conventional ASR (referred to as ASR-q). This addition is confusing and deviates from standard evaluation practices, where average number of queries and ASR are reported. The number of queries made, while insufficient [4], represents the attack efficiency, while continuing attacks after successful jailbreak attempts to exhaust the query budget, as done for ASR-a, is unnecessary and skews the assessment of attack efficiency. Relying on one of the authors as a human evaluator is also at risk of subjective bias and fairness issues. If a larger-scale human study is not possible, it is suggested that the authors report the raw ASRs as computed by the LLM-based judges. To make the evaluation more robust, the authors are also encouraged to employ more evaluation templates such as [9].

[1] Many-shot jailbreaking

[2] Autodan-turbo: A lifelong agent for strategy self-exploration to jailbreak llms.

[3] Play Guessing Game with LLM: Indirect Jailbreak Attack with Implicit Clues

[4] WordGame: Efficient & Effective LLM Jailbreak via Simultaneous Obfuscation in Query and Response.

[5] Great, Now Write an Article About That: The Crescendo Multi-Turn LLM Jailbreak Attack

[6] Improving Alignment and Robustness with Circuit Breakers

[7] SmoothLLM: Defending Large Language Models Against Jailbreaking Attacks

[8] Purple Llama: Towards open trust and safety in the new world of generative AI

[9] A StrongREJECT for Empty Jailbreaks

**Questions:**

My questions and suggestions have mostly been covered in the weaknesses that are outlined above. The authors can either improve on the dataset, baseline, defense, evaluator choices and experiment designs or explain why no changes could be made to argue for a higher rating.

Another question to answer to which might alter my judgement on novelty is: What is the reasons behind the advantage of multi-turn JSP over pseudo multi-turn and single-turn JSP? Can the authors derive an insightful interpretation by e.g. investigating the change in hidden states (norm,. probing, etc.) when inserting or replacing a handwritten assistant tag with the true assistant tag in the input?

---

### Official Review · Reviewer_EMoX · 2024-11-03

**Soundness:** 3
**Presentation:** 3
**Contribution:** 2
**Rating:** 6
**Confidence:** 3

**Summary:**

The paper proposes a simple yet effective multi-turn jailbreak strategy, Jigsaw Puzzles (JSP), to bypass the defense mechanism of many existing LLMs. JSP splits the harmful questions into pieces and asks the LLM to implicitly concatenate them and answer the harmful question in a hypothetical scenario. Experiments show that JSP is effective against existing LLMs, including Gemini-1.5-Pro and GPT-4o.

**Strengths:**

1. JSP is easy to implement and generalizable to questions in different domains. The technique of subword dividing further makes it hard to be detected.
2. Experiments show that JSP has a ≥ 50% success rate (ASR-a) even on closed-source LLMs like Gemini-1.5-Pro and GPT-4o. The difference in success rates also helps us learn more about the underlying defense mechanism behind such closed-source models, which is further discussed in the ablation study.
3. With fabricated history, JSP can break GPT-4o with a ≥ 80% success rate (ASR-a), which can be easily achieved with APIs of existing LLMs.

**Weaknesses:**

1. The defense measure tested seems to be too simple, only modifying the system prompt. The author does not show the results against defense measures like post-generation detection. Once the harmful response is generated, it should be easily detected as harmful, even with the disclaimer.
2. Once the defender knows this way of attack, it is easy to defend, like detecting the manually concatenated user input or the appearance of the JSP prompt.
3. The method works on Gemini-1.5-Pro and GPT-4o but not on Claude. I tested the same prompt on Claude, which triggers the alarm even with subwords.

**Questions:**

1. How can JSP be made more flexible? As I mentioned in the weakness, once the defender knows the form of attack, it seems hard to attack successfully.
2. Can you modify JSP to bypass Claude's defense mechanism?

**Details Of Ethics Concerns:**

Even though the authors state in the reproducibility statement that "we do not submit the code and data used, and access is granted only by submitting a form indicating the researchers’ affiliation and the intention of use," one can easily try the method by simply using the prompt and the method provided in the paper, which could lead to harmful outcomes.

---

### Note · Authors · 2024-11-25

**Comment:**

Hi everyone, we really appreciate your insightful and valuable reviews, which help us to refine and improve our work. We have decided to withdraw our submission, and will incorporate these suggestions for revising our work.

**Withdrawal Confirmation:**

I have read and agree with the venue's withdrawal policy on behalf of myself and my co-authors.